# Climate Change, Sustainable Forest Management, ICT Nexus, and the SDG 2030: A Systems Thinking Approach

**Ernesto E. Empig** [1,*], **Ahmet Sivacioğlu** [2], **Renato S. Pacaldo** [3], **Peter D. Suson** [1], **Rabby Q. Lavilles** [1], **Maria Rizalia Y. Teves** [1], **Maria Cecilia M. Ferolin** [1] and **Ruben F. Amparado, Jr.** [1,*]

1   Iligan Institute of Technology, Mindanao State University, Iligan 9200, Philippines
2   Faculty of Forestry, Kastamonu University, Kastamonu 37210, Turkey
3   Department of Forestry, Mindanao State University—Main, Marawi 9700, Philippines
*   Correspondence: ernesto.empig@g.msuiit.edu.ph (E.E.E.); ruben.amparado@g.msuiit.edu.ph (R.F.A.J.)

**Abstract:** The 2030 global agenda for sustainable development integrates social, economic, and environmental dimensions, emphasizing peace, human rights, gender equality, and women's empowerment. The SDG framework, consisting of 17 goals, 169 targets (SDTs), and 231 indicators (SDIs), forms a complex, interconnected network that necessitates extensive research. Despite prior studies on SDG interlinkages, the integration of Climate Change (CC), Sustainable Forest Management (SFM), and Information and Communication Technology (ICT), collectively known as CSI Nexus, remains underexplored. This study addresses this gap by identifying SDTs aligned with CC, SFM, and ICT (CSI) and analyzing their linkages within the SDG framework using a systems thinking approach. The objectives are to (1) investigate and identify SDTs connected with the CSI Nexus and (2) assess the significant relationship between and among CC, SFM, and ICT. The primary method involves a simplified meta-analysis and systems thinking approach incorporating content analysis, network visualization, affiliation matrix mapping, frequency distributions, and Spearman's rho correlation. Results reveal 56 SDTs directly connected within CC + SFM + ICT, 16 within CC + SFM, one within SFM + ICT, and 51 within ICT + CC. The analysis indicates CC is significantly associated with SFM, while ICT has no significant association with CC and SFM, asserting minimal influence of ICT and SFM on the SDG 2030 framework. This research provides significant insights for decision-makers and stakeholders, contributing as a science-informed guide for priority-setting, policy coherence, and decision-making supporting the 2030 Sustainable Development Goals across sectors.

**Keywords:** CSI nexus; climate change; sustainable forest management; information and communication technology; sustainable development goals; systems thinking approach





## 1. Introduction

Over the past seven years, the United Nations member states have moved to adopt the 2030 global agenda for sustainable development published on 25 September 2015 [1], comprising 17 goals and 169 targets [2] in response to pressing concerns surrounding global sustainability. The fundamental nature SDG framework represents an indivisible and integrated network that balances the threet pillars of sustainable development [3–5]. The integration of Climate Change (CC), Sustainable Forest Management (SFM), and ICT, collectively known as CSI Nexus, are explicitly included as one goal in SDG 13 and as selected targets in SDG 15. In contrast, ICT is implicitly distributed as targets for other goals, acting as an enabler to accelerate progress in achieving the SDGs for global sustainability. The complex and interconnected nature of the SDGs demands a systems thinking approach to understand their relationships effectively [6]. Previous research has used various tools and methodologies to explore SDG interconnections and integration of related frameworks [4,5,7–10].

CC affects 85% of the world's population and disproportionately affects vulnerable groups [11–14]. Climate change may cause extreme weather, poverty, and poor health,

making it critical to reaching the SDGs. Current warming trends may cause temperatures to rise by 1.5 °C in the next 20 years. SFM is crucial for mitigating climate change, conserving biodiversity, and providing ecosystem services [15–17]. However, deforestation and unsustainable practices continue to threaten these benefits [18].

Information and Communication Technology (ICT) enables global sustainability by offering innovative solutions for environmental monitoring, resource management, and sustainable practices, accelerating progress in areas such as education, healthcare, and economic opportunities [19–25]. The natural relationship between CC and forests, and the potential of ICT to prevent climate change and enhance SFM, emphasizes the interconnectedness of these domains in achieving global sustainability. A systems thinking approach is necessary to understand the interlinkages and interactions among SDGs, CC, SFM, and ICT. Although various research has investigated these links, a thorough understanding of their alignment with all 169 SDG targets and the role of ICT as an SDG catalyst remains a topic for further investigation, for there is a scarcity of literature aligning the SDTs into CC, SFM, and ICT [22,24–33].

Moreover, the complexity of the SDGs demands systems thinking approach to analyze the intricate interconnections among the 169 SDG targets [34–37]. Systems thinking, a set of synergistic analytical skills, has gained prominence in sustainable development research due to the integrated nature of the SDGs [38]. Recent studies have employed systems thinking in various aspects of SDGs, including assessing Nigeria's floods and food security [39], proposing an 18th SDG for digital connection [40], and strengthening aquaculture policy [41]. This study utilizes the systems thinking approach to analyze and visualize the interlinkages among CC, SFM, ICT, and the SDG targets. Although numerous studies have explored the connections between SDGs, CC, SFM, and ICT [42–45], substantial gaps persist in examining their interlinkages and interactions [46]. The complexity of the SDGs' network poses challenges for signatory countries, requiring further research [34]. Therefore, to our knowledge, no existing study has explicitly integrated CC, SFM, and ICT within the Framework of SDG 2030.

Generally, understanding the interconnection of SDTs, CC, SFM, and ICT is essential for developing global sustainability strategies. Addressing climate change, managing forests responsibly, and using ICT is crucial since these fields are interconnected. We can build a more resilient, equitable, and sustainable future by addressing these issues holistically. This comprehensive approach to global sustainability ensures informed policies and initiatives for a better future.

In response to these global challenges, this study presents a novel approach and is a pioneering study that addresses the gaps in understanding the interrelationships among CC, SFM, and ICT in the context of the SDGs. This study aims to (1) investigate and identify relevant SDG targets connected to CSI Nexus; and (2) assess the significant relationships between and among CC, SFM, and ICT. Additionally, this study hypothesizes that there are no significant associations between these three domains within the context of the SDGs. To achieve these objectives, this study utilizes a primary methodology that combines simplified meta-data and systems thinking techniques, such as content analysis, network visualization, affiliation matrix mapping, frequency distributions, and Spearman's rho correlation. These methodologies enable a comprehensive understanding of the interconnectedness within the SDGs, facilitating the identification of key targets, areas for further research and action, and the development of effective strategies to address global sustainability challenges. The resulting CSI Nexus and SDT Integration Framework guide science-informed decision-making, policy coherence, priority setting, and practical resource allocation for project funding and implementation, ultimately addressing global sustainability challenges.

## 2. Materials and Methods

The study employs a combined methodology of simplified meta-data analysis and systems thinking to identify specific SDG targets related to Climate Change (CC), Sustainable Forest Management (SFM), and Information and Communication Technology (ICT) and

assess their significant relationships within the SDG context. This method includes data collection, content analysis, affiliation matrix mapping, frequency distributions, network analysis and visualization, and Spearman's rho correlation. It allows the development of the CSI Nexus and SDTs Integration Framework, which helps decision-makers, researchers, and practitioners develop effective, integrated strategies to address global sustainability challenges and reach the SDGs by 2030 [46,47].

The analytical Framework (Figure 1) integrates CC, SFM, ICT, and the SDG Framework from a siloed structure to an integrated and systemic approach, guiding the methodology in each of the three phases. These phases include the following: (1) identification of SDTs aligned with CC, SFM, and ICT through a meta-data analysis of existing literature; (2) mapping and integration of the identified SDTs to CC, SFM, and ICT using a systems thinking approach to analyze interlinkages among the SDTs; and (3) quantification of the integrated CSI Nexus and SDTs using statistical techniques such as correlation analysis and network analysis. The Framework used to integrate CC, SFM, ICT, and the SDG Framework into a systemic approach, guiding the methodology in each of the three phases to ensure a structured approach focused on relevant SDTs and their alignment with CC, SFM, and ICT.

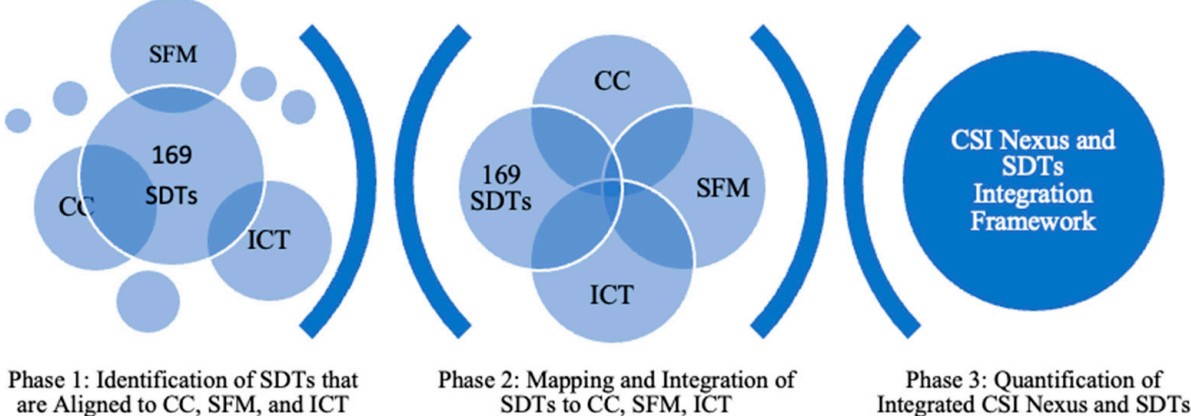

**Figure 1.** Analytical Framework for Integrating CC, SFM, ICT, and the SDTs.

*2.1. Phase 1: Identification of SDTs That Are Aligned to CC, SFM, and ICT*

This phase involved an intensive review of literature and data sources in determining relevant SDTs contributing to greenhouse gas emissions [32], sustainable forest management practices, and the use of ICT in achieving the Sustainable Development Goals (SDGs) [48]. Phase 1 aimed to identify SDTs aligned with CC, SFM, and ICT. It also aimed to evaluate the identified SDTs across CC, SFM, and ICT and to illustrate the interlinkages among the identified SDTs and the three CSI Nexus domains using a circular relationship diagram. Additionally, in this Phase, the 169 SDTs were connected to CC, SFM, and ICT (CSI) separately.

2.1.1. Data Collection and Identification of SDTs

This task involved an extensive review of literature, reports, and databases related to SDGs, CC, SFM, and ICT in determining their alignment with the three CSI domains. The study utilized twelve sources, comprising four articles on climate change [10,29,32,49], four on sustainable forest management [24,31,33,50], and four on ICT [48,51–53]. A combination of keywords, such as "Sustainable Development Goals", "Climate Change", "Forest", "ICT", "mapping", "integration", "connect", and "interlink", was used in various research databases like Google Scholar, Google Search, Science Direct, MDPI, Scopus, Web of Science, Directory of Open Access Journal, and Microsoft Academic.

The literature was imported into the Reference Management Software EndNote 20 to filter out duplicates and irrelevant materials. Due to the scarcity of literature, the search was expanded to include reports and documents from renowned international organizations,

such as the United Nations, Food and Agriculture Organizations, Huawei, Ericson, Forest Stewardship Council, World Summit on the Information Society, Intergovernmental Panel on Climate Change, and International Telecommunication Union. The selected twelve sources were reviewed to identify SDTs aligned with CC, SFM, and ICT.

The SDTs aligned with the CSI Nexus domains were identified, tallied, and categorized into four groups: SDGs, CC, SFM, and ICT. Table 1 displays examples of SDTs aligned with CC, SFM, and ICT, along with their corresponding references and the specific SDTs contributing to the achievements of SDGs. The clustering of SDTs and the CSI Nexus domains was illustrated using a circular relationship diagram in phase 1 (Figure 1), which visually represents the interlinkages among the SDTs, and the three groups of data sources identified in the study. The diagram demonstrates the unequal overlap of the three domains within the circle of SDTs, with the CC circle containing the most SDTs, followed by ICT and then SFM.

**Table 1.** List of Connected SDTs as a Data Source Aligned to CC, SFM, and ICT.

| CSI Clusters Code | Literature References | Connected SDG Targets (SDTs) |
|---|---|---|
| Climate Change (CC1) | Nerini, F. F; Sovacool, B.; Hughes, N.; Cozzi, L.; Cosgrave, E.; Howells, M.; Tavoni, M.; Tomei, J.; Zerriffi, H.; Milligan, B. Connecting climate action with other Sustainable Development Goals. Nature Sustainability 2019, 2, 674–680. https://doi.org/10.1038/s41893-019-0334-y [10]. | 1.1, 1.2, 1.3, 1.4, 1.5, 1.b, 2.1, 2.2, 2.3, 2.4, 2.5, 2.c, 3.1, 3.2, 3.3, 3.4, 4.1, 4.2, 4.5, 5.1, 5.2, 5.5, 5.a, 5.b, 5.c, 6.1, 6.2, 6.4, 6.6, 7.1, 7.2, 7.b, 8.1, 8.3, 8.4, 8.5, 8.6, 8.8, 8.9, 8.10, 9.1, 10.1, 10.2, 10.6, 10.7, 11.1, 11.2, 11.5, 12.1, 12.2, 13.1, 13.2, 13.3, 13.a, 13.b, 14.1, 14.2, 14.3, 14.4, 14.7, 14.a, 14.b, 15.1, 15.2, 15.3, 15.4, 15.5, 15.6, 15.8, 15.c, 16.1, 16.7. |
| Sustainable Forest Management (SFM1) | FAO. 2018. The State of the World's Forests 2018—Forest pathways to sustainable development. Rome. License: CC BY-NC-SA 3.0 IGO. Available at https://www.fao.org/3/I9535EN/i9535en.pdf accessed on 5 February 2023 [51]. | 1.1, 1.4, 1.5, 2.1, 2.3, 5.5, 5.a, 6.6, 6.6.1, 7.1, 7.2, 8.3, 8.9, 11.4, 11.7, 12.2, 12.5, 12.6, 12.7, 13.1, 13.2, 13.3, 15.1, 15.1.1, 15.1.2, 15.2, 15.2.1, 15.3, 15.4, 15.4.1, 15.4.2, 15.5, 15.5.1, 15.b |
| Information and Communication Technology (ICT1) | Partnership on Measuring ICT for Development, 2019. A thematic list of ICT indicators for the SDG. Available at https://www.itu.int/en/ITUD/Statistics/Documents/intlcoop/partnership/Thematic_ICT_indicators_for_the_SDGs.pdf accessed on 5 February 2023 [52] | 1.4, 2.3, 2.a, 2.c, 3.8, 4.4, 4.5, 4.a, 5.b, 8.1, 8.2, 8.3, 8.5, 8.10, 9.1, 9.5, 9.a, 9.c, 10.c, 12.4, 12.5, 12.8, 16.6, 16.7, 16.10, 17.6, 17.8 |

Note: Refer to Table S1 of Supplementary Materials for complete list of connected SDTs as data source aligned to CC, SFM, and ICT.

### 2.1.2. Clustering of Literature Sources Connecting SDTs to CC, SFM, and ICT

Table 2 lists the twelve literature sources used as data sources for the CSI Nexus. The table is organized into three columns, representing each of the three domains. It lists the corresponding literature sources, their codes, and the number of Sustainable Development Goals (SDGs) and Sustainable Development Technologies (SDTs) they address. The clustering of literature sources connecting SDTs to CC, SFM, and ICT was based on a content analysis of the identified sources.

The twelve literature sources were screened into four categories: the 169 SDTs [1,54], Climate Change, Sustainable Forest Management, and Information and Communication Technology. The total number of goals and targets of SDGs in each domain was summarized to determine the contribution of CSI to the SDG framework as a whole. The literature sources were selected based on their alignment with the focal areas of CC, SFM, and ICT, regardless of either positive or negative linkages. Each literature source was reviewed and analyzed for its connections to the three domains. The content analysis-based clustering of literature sources provided a foundation for developing the analytical Framework and subsequent phases of the methodology.

**Table 2.** List of twelve literature as data sources of CC, SFM, and ICT integration.

| Climate Change (CC) | | | Sustainable Forest Management (SFM) | | | Sustainable Forest Management (SFM) | | |
|---|---|---|---|---|---|---|---|---|
| Code | Literature | SDGs & SDTs | Code | Literature | SDGs & SDTs | Code | Literature | SDGs & SDTs |
| CC1 | Nerini, F. et al., 2019 [10] | 16 & 72 | SFM1 | FAO, 2018 [51] | 11 & 28 | ICT1 | PMID, 2019 [52] | 11 & 27 |
| CC2 | IPCC, 2018 [32] | 17 & 121 | SFM2 | OLI-UNFF, 2016 [24] | 8 & 17 | ICT2 | WSIS 2015 [53] | 17 & 83 |
| CC3 | FAO, 2019 [29] | 17 & 89 | SFM3 | FSC, 2019 [31] | 14 & 40 | ICT3 | Huawei, 2018 [54] | 6 & 55 |
| CC4 | Zhou, X. et al., 2021 [50] | 17 & 131 | SFM4 | WBCSD-FSG, 2019 [33] | 11 & 43 | ICT4 | Ericsson & EICU, 2017 [25] | 9 & 27 |

### 2.2. *Phase 2: Mapping and Integration of SDTs to CC, SFM, and ICT*

Phase 2 focuses on mapping and integrating SDT) with CC, SFM, and ICT. This process employs a Venn diagram approach, an affiliation matrix, network analysis, and visualization to identify potential leverage points for addressing sustainability challenges. To achieve a comprehensive understanding of the relationships among SDTs, CC, SFM, and ICT, this Phase has three primary objectives: (1) Merge the four groups (CC, SFM, ICT, and SDTs) using a Venn diagram to represent the interconnections among the three domains and SDTs visually; (2) Represent relationships using an affiliation matrix to analyze the relationships between SDTs and the CSI, revealing direct connections and overlaps; and (3) Perform network analysis and visualization using a systems thinking approach to analyze complex interconnections among the CSI and SDTs and visualize the network to identify potential leverage points and areas for further research and action.

### 2.2.1. Venn Diagram Approach and Clustering

The Venn diagram approach is a visual tool demonstrating the complex interconnections among these focal areas and SDTs. It allows researchers to thoroughly understand their relationships and identify potential leverage points for addressing sustainability challenges. The methodology involves creating a Venn diagram with three overlapping circles representing CC, SFM, and ICT. The overlaps indicate the intersection points and relationships among the groups. The SDTs are then clustered into seven categories based on their interconnections with the focal areas: CC, SFM, ICT, CC + SFM, SFM + ICT, ICT + CC, and CC + SFM + ICT. This clustering process clarifies the areas of overlap and distinction between the focal areas and SDTs, providing valuable insights into their relationships, as depicted in Figure 2.

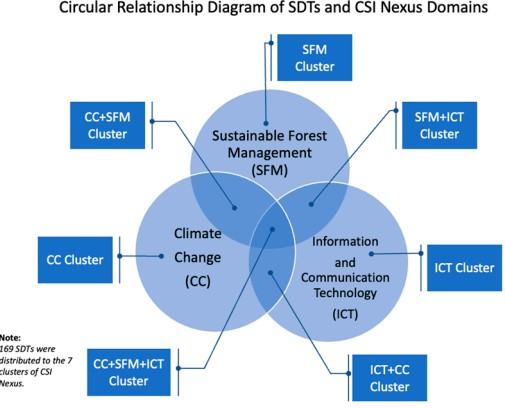

**Figure 2.** Venn Diagram Illustrating the Interconnections and Clustering of SDTs with CSI.

#### 2.2.2. A Systems Thinking Approach in Mapping SDTs Using Affiliation Matrix

This task uses a systems thinking approach [40] and an affiliation matrix [55] to holistically analyze the interconnections between SDTs and CSI. Applying systems thinking principles, such as interconnectedness, emergent properties, system boundaries [56], and the affiliation matrix visually represents these relationships, revealing patterns, overlaps, gaps, and emergent properties. This enhanced mapping offers valuable insights into the complex relationships between SDTs and CSI, informing policy development and effective interventions for sustainability challenges.

The affiliation matrix technique visually maps and analyzes SDTs' interconnections with the focal areas using a two-mode rectangular network containing actor and event nodes [57]. The matrix identifies clusters of interconnected SDTs, potential leverage points for addressing sustainability challenges, and areas for further research and action [56]. SDTs and CSI are listed as rows and columns in the matrix, with "1" indicating a relationship and "0" indicating no relationship.

Table 3 presents the affiliation matrix with a binary notation indicating the relationship between specific SDTs and CSI. The total number of SDTs aligned with each focal area is provided, and the SDTs are clustered into four categories based on their interconnections with the focal areas, including CC + SFM, SFM + ICT, ICT + CC, and CC + SFM + ICT. This visualization method helps identify patterns, overlaps, and gaps in the relationships between SDTs and focal areas, identifying potential leverage points for addressing sustainability challenges and areas for further research and action. In the matrix, 0 means not connected, 1 means connected. However, in the total connections of each domain of CC, SFM, and ICT is 1 to 4, as indicated in the respective Total column. The possible total of combining the three domains can be 1 to 12, as indicated in the CC + SFM + ICT Total column.

**Table 3.** Example of Affiliation Matrix Mapping: Linkages of SDTs to CC, SFM, and ICT.

| 17 SDGs | 169 Targets | Climate Change Mitigation and Adaptation (CC) | | | | | Sustainable Forest Management (SFM) | | | | | Information & Communication Technology (ICT) | | | | | CC + SFM + ICT (CSI) Total | | | | | CC + SFM + ICT Integration | | | | | | |
|---|---|---|---|---|---|---|---|---|---|---|---|---|---|---|---|---|---|---|---|---|---|---|---|---|---|---|---|---|
| | | Nerini, F. 2019 [10] | IPCC., 2019 [32] | FAO, 2019 [29] | Zhou, X., et al., 2021 [50] | Total (CC) | FAO, 2018 [51] | OLI-UNFF, 2016 [24] | FSC, 2019 [31] | WBCSD-FSG, 2019 [33] | Total (SFM) | PMID, 2019 [52] | WSIS, 2015 [53] | Huawei, 2018 [54] | ERICSSON & EICU, 2017 [25] | Total (ICT) | CC | SFM | ICT | Total (CSI) | CC + SFM + ICT | CC + SFM | CC + ICT | SFM + ICT | CC | SFM | ICT | No Connection |
| Goal 1: No Poverty | 1.1 | 1 | 1 | 1 | 1 | 4 | 1 | 0 | 0 | 0 | 1 | 0 | 0 | 0 | 0 | 0 | 4 | 1 | 0 | 5 | 0 | 1 | 0 | 0 | 0 | 0 | 0 | 0 |
| | 1.2 | 1 | 1 | 1 | 1 | 4 | 0 | 0 | 0 | 0 | 0 | 0 | 0 | 0 | 0 | 0 | 4 | 0 | 0 | 4 | 0 | 0 | 0 | 0 | 1 | 0 | 0 | 0 |
| | 1.3 | 1 | 1 | 1 | 1 | 4 | 0 | 0 | 0 | 0 | 0 | 0 | 0 | 0 | 0 | 0 | 4 | 0 | 0 | 4 | 0 | 0 | 0 | 0 | 1 | 0 | 0 | 0 |
| | 1.4 | 1 | 1 | 1 | 1 | 4 | 1 | 0 | 0 | 1 | 2 | 1 | 1 | 0 | 1 | 3 | 4 | 2 | 3 | 9 | 1 | 0 | 0 | 0 | 0 | 0 | 0 | 0 |
| | 1.5 | 1 | 1 | 1 | 1 | 4 | 1 | 0 | 1 | 1 | 3 | 0 | 1 | 0 | 0 | 1 | 4 | 3 | 1 | 8 | 1 | 0 | 0 | 0 | 0 | 0 | 0 | 0 |
| | 1.a | 0 | 1 | 1 | 1 | 3 | 0 | 0 | 0 | 0 | 0 | 0 | 0 | 0 | 0 | 0 | 3 | 0 | 0 | 3 | 0 | 0 | 0 | 0 | 1 | 0 | 0 | 0 |
| | 1.b | 1 | 1 | 1 | 1 | 4 | 0 | 0 | 0 | 0 | 0 | 0 | 1 | 0 | 0 | 1 | 4 | 0 | 1 | 5 | 0 | 0 | 1 | 0 | 0 | 0 | 0 | 0 |

#### 2.2.3. Network Visualization Using Systems Thinking Approach

This methodology employs the systems thinking approach to visualize and analyze the network connections between SDTs and the CSI. The method, emphasized by Pereira et al. and Laumann et al., is beneficial for understanding the intricate network of interlinkages between the SDGs, given their universal and integrated nature [34,35].

The first step in the methodology involves importing the affiliation matrix into Gephi®, an open-source network analysis and visualization software. This software allows researchers to generate a graphical representation of the network and observe the connections

and interactions between SDTs and focal areas. Direct alignment of SDTs with the CC, SFM, and ICT (CSI) nexus was performed using the data from twelve selected publications. A preferential attachment network is employed to analyze the interlinkages between these domains further. This method facilitates thematic clustering of the interlinkages among CC, SFM, and ICT, providing a more comprehensive understanding of their relationships.

Figure 3a shows a complex network visualization based on twelve data sources (CC1 to CC4, SFM1 to SFM4, and ICT1 to ICT4), comprising 181 nodes and 733 edges. The nodes in this visualization are color-coded based on their clusters. Pink nodes indicate the intersection of CC, SFM, and ICT; blue nodes represent CC; orange nodes represent ICT; purple nodes denote the overlap between CC and SFM; yellow nodes signify the connection between SFM and ICT; red nodes indicate the link between ICT and CC; dark blue, dark green, and dark orange nodes represent CC, SFM, and ICT hubs, respectively, and gray corresponds to nodes with no connection. To simplify the network visualization, we merged the four similar hubs into a single network, resulting in an undirected graph with 155 nodes and 332 edges. This integrated network system was then organized into three main hubs: CC, SFM, and ICT. We removed all SDTs with only zero or one interconnection in Figure 3b. This streamlined visualization allows for a more straightforward interpretation and understanding of the relationships between the focal areas and SDTs. The colors used in Figure 3b are consistent with those used in Figure 3a. The CC-hub node is represented in dark blue, SFM-hub node in dark green, ICT-hub node in dark orange, CC nodes in blue, purple nodes interconnect CC and SFM, yellow node interconnects SFM and ICT, orange nodes in ICT, red nodes interconnect ICT and CC, and pink nodes interconnect CC, SFM, and ICT.

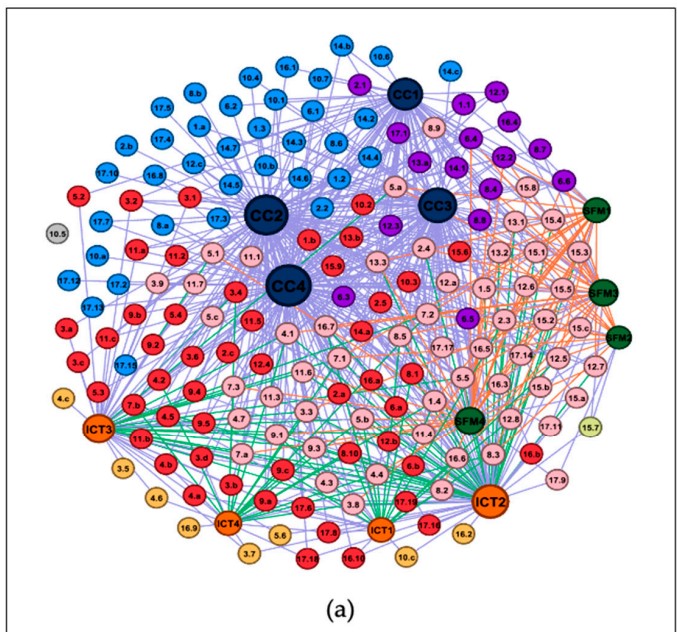 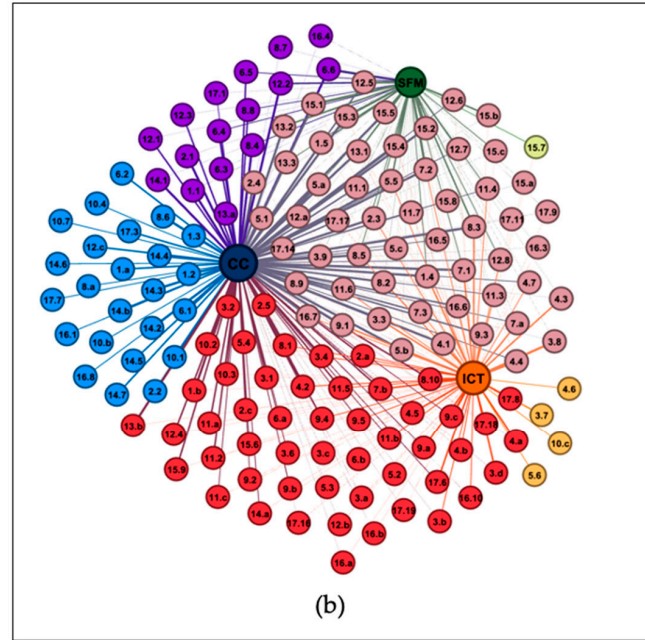

**Figure 3.** Simplified Network Visualization of Interconnection Between SDTs and CSI Nexus.

Refinement of the network visualization helps understand the network's structure and links between SDTs and focal areas. Clustering methods may be used to discover similar groups, filter the network to focus on specific connections or nodes, and change the layout to highlight attributes such as node centrality or modularity. These enhancements allow for a more extensive analysis of component interactions, identifying relevant policy interventions and areas for further research. It ultimately leads to better-informed policy development and interventions.

### 2.3. Phase 3: Quantification of the Relationships between and among CC, SFM, and ICT

In this phase, the objectives are to investigate relevant SDG targets and assess the significant relationships among CC, SFM, and ICT. This phase employs a comprehensive methodology integrating frequency distributions, network analysis focusing on degree centrality, and Spearman's rho correlation. A hypothesis posits no significant associations between them. This approach facilitates an understanding of SDG interconnectedness. The method for Phase 3 includes the following steps:

Step 1: Network Analysis—Degree Centrality

Degree centrality measures connections within the network, providing insights into the importance and potential influence of each SDT. It employs network analysis to examine relationships among SDTs in the context of CC, SFM, and ICT, focusing on degree centrality to identify central and influential SDTs within the network. The network analysis process includes data preparation, software selection, degree centrality calculation, and interpretation.

Step 2: Frequency Distributions Analysis

It involves frequency distribution analysis to determine the occurrence of each SDT in the context of CC, SFM, and ICT. The research identifies the most relevant SDTs, informing policy development and interventions. A dataset was created for frequency distribution analysis, which was then performed using the statistical software Jamovi.

Step 3: Spearman's Rho Correlation Analysis

It employs Spearman's rho correlation analysis to determine the strength and direction of relationships between CC, SFM, ICT, and SDTs. Using Jamovi, correlation coefficients are generated, highlighting unique patterns and relationships. This step allows researchers to understand SDG interconnectedness better, guiding policy and initiatives.

Step 4: Hypothesis Testing

It tests null and alternative hypotheses by analyzing CC, SFM, and ICT relationships. Appropriate statistical tests are chosen based on data and research objectives, and the data from previous phases are prepared for hypothesis testing. Using Jamovi, the selected tests examine the relevance of correlations between CC, SFM, ICT, and SDTs. The test outcomes help researchers answer study questions and objectives, determining the importance of relationships and informing policy responses.

### 2.4. Social Network Analysis Using Gephi

Social network analysis helps identify central nodes in network systems through centrality measures such as degree, distance, betweenness, and eigenvector. This study uses these centrality metrics to determine the most significant targets in CSI-Nexus and SDTs integration. High degree centrality indicates broader interactions, while high closeness centrality suggests a node is closer to all network targets. High betweenness centrality implies a node's considerable influence due to its control over information passing between others. Eigenvector centrality accounts for both the number of neighbors and their significance.

The data were collected, plotted in the affiliation matrix, and input into a spreadsheet as nodes (targets) and edges (connections). Two spreadsheets were prepared to build the network for analysis: one for nodes with ID, Label, and other information and another for edges with Source, Target, Type, and Weight. Both files were imported into Social Network Analysis software like Gephi® for analysis and visualization. The network was simulated based on desired output appearance and network statistics. The data laboratory displayed the results of network centralities, such as degree centrality, distance, betweenness, and eigenvector, as shown in Figure 4.

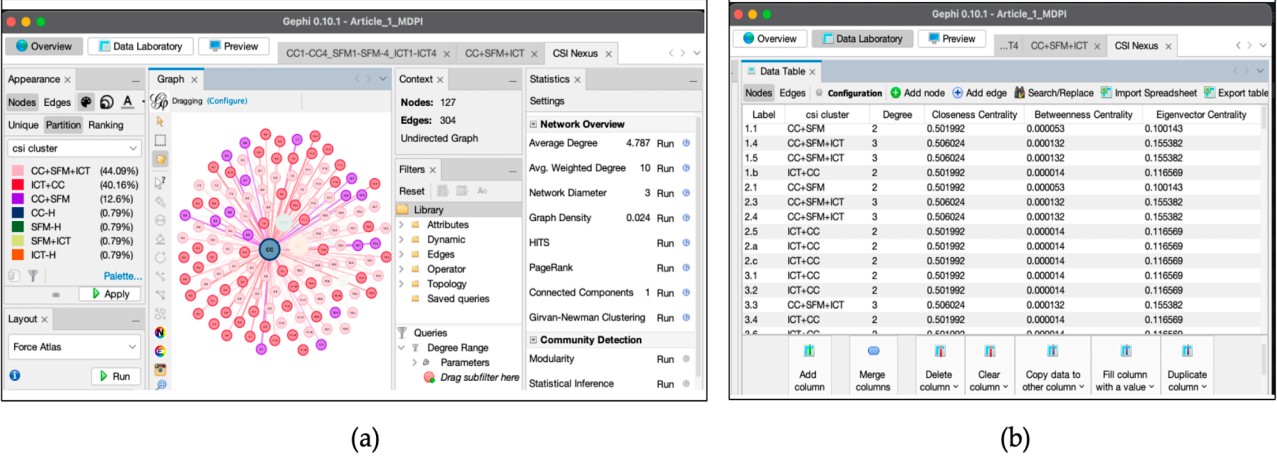

<div style="text-align: center;">

(a)           (b)

</div>

**Figure 4.** Illustration of Network Analysis and Visualization of the Interconnections of CSI and SDTs using Gephi: (**a**) Network visualization of CSI-SDT Integration; (**b**) Network Analysis Showing the data table of Network Centrality.

## 3. Results

In this results section, we present our findings related to identifying relevant SDG targets connected to the CSI Nexus and assessing significant relationships between CC, SFM, and ICT. Additionally, we examine the hypothesis that there are no significant associations between these three domains within the context of the SDGs. The insights obtained contribute to developing effective strategies for addressing global sustainability challenges.

### 3.1. Identifying and Analyzing SDG Targets Related to the CSI Nexus

Figure 5 presents the result of the distribution of SDTs to CSI clusters. The pie chart of CSI Nexus was categorized into seven groups based on their interconnections: No Connection, CC, ICT, CC + SFM, SFM + ICT, ICT + CC, and CC + SFM + ICT. The largest cluster is CC + SFM + ICT, with 33.1% of the total SDTs, followed by CC + ICT, with 30.2%. CC alone has 21.3%, while SFM + ICT and No Connection have the lowest percentage with only 0.6% each. The pie chart provides a clear visual representation of the distribution of SDTs across the different clusters. It highlights the prominent role of CC, SFM, and ICT in achieving the SDGs.

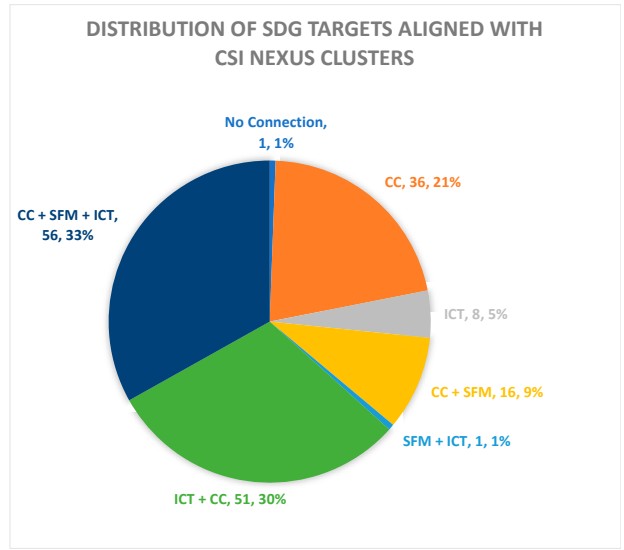

**Figure 5.** Pie Chart Distribution of SDTs Aligned with CSI Nexus Clusters.

### 3.2. Clustering and Interlinkages of SDTs in the CSI Nexus

Figure 6 shows the clustered literature sources that connect SDTs to the three domains and summarizes the findings in a bar chart. The x-axis shows the different sources of the three domains, from CC1-4, SFM1-4, and ICT1-4. The results showed that CC has the highest number of SDTs, with CC4 having the most literature references at 131 and a percentage equivalent of 78%. SFM3 and SFM4 followed with 43 and 40 SDTs, respectively, with SFM4 having the highest percentage equivalent of 25%. For ICT, ICT2 had the highest number of SDTs, with 83, and a percentage equivalent of 49%. The clustering of literature sources highlights the significant role of CC, SFM, and ICT in achieving the SDGs.

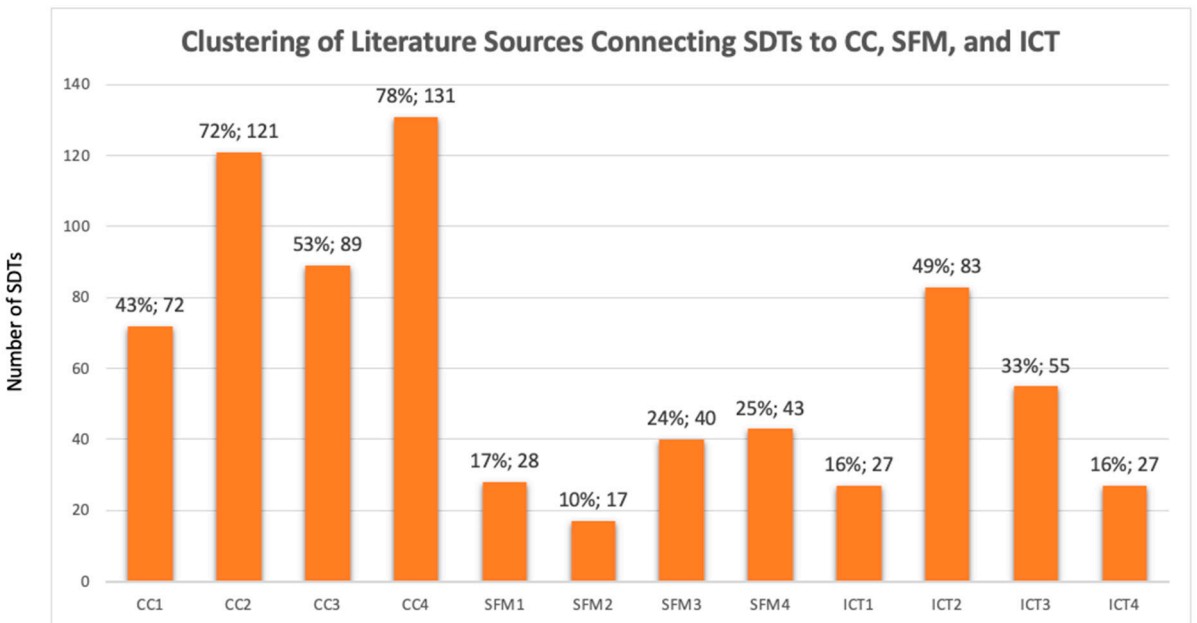

**Figure 6.** Bar Chart for Clustering of Literature Sources connecting SDTs to CSI Domains. Note: CC1—Nerini et al., 2019; CC2—Intergovernmental Panel for Climate Change (IPCC), 2018; CC3—Food and Agriculture Organization, 2018; CC4—Zhou et al., 2019; SFM1—Food and Agriculture Organization, 2018; SFM2—Organization-led Initiative in support of the United Nations Forum on the Forest, 2017; SFM3—Forest Stewardship Council, 2019; SFM4—World Business Council for Sustainable Development, 2019; ICT1—Partnership on Measuring ICT for Developments, 2019; ICT2—International Telecommunications Union—World Summit on Information Sciences, 2015; ICT3—Huawei, 2018; ICT4—The Earth Institute Columbia University and Ericsson, 2017.

Additionally, this presents the results of clustering and interlinkages of sustainable development targets (SDTs) within the CSI Nexus, which is categorized into seven groups based on the degree of connection between SDTs and three CSI domains, namely CC, SFM, and ICT. To illustrate the overlapping among SDTs and the three CSI Nexus domains, the researchers provided a circular relationship diagram using the Venn diagram (Figure 2). The chart shows the seven CSI Nexus clusters: CC, SFM, ICT, CC + SFM, SFM + ICT, ICT + CC, and CC + SFM + ICT.

Table 4 presents the distributed SDTs of each CSI Cluster and their strength of connections based on the number of sources connected in each SDT. The scale range used for the connection strength was 1–4 for weak connections, 5–8 for moderate connections, 9–12 for strong connections, and 0 for no connections.

**Table 4.** Frequency Distributions of SDTs to the CSI Nexus Clusters.

| CSI Clusters | Counts | Percentage | Strength of Connection | | | |
|---|---|---|---|---|---|---|
| | | | Weak Connection | Moderate Connection | Strong Connection | No Connection |
| No Connection | 1 | 1% | | | | 10.5 |
| CC | 36 | 21% | 1.2, 1.3, 1.a, 2.2, 2.b, 6.1, 6.2, 8.6, 8.a, 8.b, 10.1, 10.4, 10.6, 10.7, 10.a, 10.b, 12.c, 14.2, 14.3, 14.4, 14.5, 14.6, 14.7, 14.b, 14.c, 16.1, 16.8, 17.2, 17.3, 17.4, 17.5, 17.7, 17.1, 17.12, 17.13, 17.15 | | | |
| ICT | 8 | 5% | 3.5, 3.7, 4.6, 4.c, 5.6, 10.c, 16.2, 16.9 | | | |
| CC + SFM | 16 | 9% | 6.3, 8.7, 12.1, 12.3, 16.4, 17.1 | 1.1, 2.1, 6.4, 6.5, 6.6, 8.4, 8.8, 12.2, 13.a, 14.1 | | |
| SFM + ICT | 1 | 1% | 15.7 | | | |
| ICT + CC | 51 | 30% | 2.5, 2.c, 3.1, 3.6, 3.a, 3.b, 3.c, 3.d, 4.a, 4.b, 5.2, 5.3, 5.4, 6.a, 6.b, 9.2, 9.5, 9.b, 10.3, 11.2, 11.a, 11.b, 11.c, 12.4, 12.b, 14.a, 15.6, 15.9, 16.10, 16.a, 16.b, 17.8, 17.16, 17.18, 17.19 | 1.b, 2.a, 3.2, 3.4, 4.2, 4.5, 7.b, 8.1, 8.10, 9.4, 9.a, 9.c, 10.2, 11.5, 13.b, 17.6 | | |
| CC + SFM + ICT | 56 | 33% | 5.c, 11.7, 15.8, 15.a, 15.c, 16.3, 16.5, 17.9, 17.11 | 1.5, 2.3, 2.4, 3.3, 3.8, 3.9, 4.1, 4.3, 4.4, 4.7, 5.1, 5.a, 7.3, 7.a, 8.2, 8.5, 8.9, 9.1, 9.3, 11.1, 11.3, 11.4, 11.6, 12.5, 12.6, 12.7, 12.8, 12.a, 13.1, 13.2, 13.3, 15.4, 15.b, 16.6, 16.7, 17.14, 17.17 | 1.4, 5.5, 5.b, 7.1, 7.2, 8.3, 15.1, 15.2, 15.3, 15.5 | |

The table provides each cluster's counts and percentage distributions and the specific SDTs with their connection strength to each CSI domain. Most SDTs are connected to CC, followed by ICT and SFM. The first cluster, "No Connection", has only one SDT that does not link to any of the three CSI domains. The second cluster, CC, has 36 SDTs with a moderate to strong connection to the CC domain. The third cluster, ICT, has eight SDTs with a weak to moderate connection to the ICT domain. The fourth cluster, CC + SFM, has 16 SDTs with a moderate to strong connection to CC and SFM domains. The fifth cluster, SFM + ICT, has one SDT with a moderate connection to the SFM domain and a weak connection to the ICT domain. The sixth cluster, ICT + CC, has 51 SDTs that have a weak to strong connection to both ICT and CC domains. The last cluster, CC + SFM + ICT, has 56 SDTs that have a moderate to strong connection to all three CSI domains.

Overall, the Venn diagram (Figure 2) and Table 4 clearly represent the interconnections among the SDTs, and the three groups of data sources identified in the study.

### 3.3. Systems Thinking Approach in Mapping SDTs to Heat Map Affiliation Matrix

Table 5 presents a heat map affiliation matrix that shows the top significant SDTs to CSI domains and clusters, highlighting the systems thinking approach through visual representation. The matrix is sorted based on the highest CSI values, and the top SDTs have been identified. The CSI score is based on the targets' performance in three categories: CC, SFM, and ICT. The highest scoring targets are Target 7.2 and 8.3, with a CSI score of 10, indicating that they are strongly connected in all three domains.

**Table 5.** Heat Map Affiliation Matrix of Significant SDTs to CSI Domains and Clusters.

| SDG Targets | CC | SFM | ICT | CSI Total | CSI Clusters | Rank Average |
|---|---|---|---|---|---|---|
| Target 7.2 | 4 | 4 | 2 | 10 | CC + SFM + ICT | 1.5 |
| Target 8.3 | 4 | 3 | 3 | 10 | CC + SFM + ICT | 1.5 |
| Target 1.4 | 4 | 2 | 3 | 9 | CC + SFM + ICT | 6.5 |
| Target 5.5 | 4 | 3 | 2 | 9 | CC + SFM + ICT | 6.5 |
| Target 5.b | 4 | 1 | 4 | 9 | CC + SFM + ICT | 6.5 |
| Target 7.1 | 4 | 2 | 3 | 9 | CC + SFM + ICT | 6.5 |
| Target 15.1 | 4 | 4 | 1 | 9 | CC + SFM + ICT | 6.5 |
| Target 15.2 | 4 | 4 | 1 | 9 | CC + SFM + ICT | 6.5 |
| Target 15.3 | 4 | 4 | 1 | 9 | CC + SFM + ICT | 6.5 |
| Target 15.5 | 4 | 4 | 1 | 9 | CC + SFM + ICT | 6.5 |
| Target 1.5 | 4 | 3 | 1 | 8 | CC + SFM + ICT | 16 |
| Target 2.3 | 4 | 2 | 2 | 8 | CC + SFM + ICT | 16 |
| Target 3.3 | 4 | 1 | 3 | 8 | CC + SFM + ICT | 16 |
| Target 4.1 | 4 | 1 | 3 | 8 | CC + SFM + ICT | 16 |
| Target 6.6 | 4 | 4 | 0 | 8 | CC + SFM | 16 |
| Target 8.5 | 4 | 2 | 2 | 8 | CC + SFM + ICT | 16 |
| Target 9.1 | 4 | 1 | 3 | 8 | CC + SFM + ICT | 16 |
| Target 12.5 | 3 | 4 | 1 | 8 | CC + SFM + ICT | 16 |
| Target 13.1 | 4 | 3 | 1 | 8 | CC + SFM + ICT | 16 |
| Target 13.2 | 4 | 3 | 1 | 8 | CC + SFM + ICT | 16 |
| Target 15.4 | 4 | 3 | 1 | 8 | CC + SFM + ICT | 16 |
| Target 4.4 | 2 | 1 | 4 | 7 | CC + SFM + ICT | 24.5 |
| Target 5.a | 4 | 2 | 1 | 7 | CC + SFM + ICT | 24.5 |
| Target 9.3 | 3 | 1 | 3 | 7 | CC + SFM + ICT | 24.5 |
| Target 12.2 | 4 | 3 | 0 | 7 | CC + SFM | 24.5 |
| Target 13.3 | 4 | 2 | 1 | 7 | CC + SFM + ICT | 24.5 |
| Target 16.7 | 4 | 1 | 2 | 7 | CC + SFM + ICT | 24.5 |

Targets connected in the three domains are assigned to the CSI Cluster CC + SFM + ICT, while those related to two domains but not in the third are assigned to the CSI Clusters of CC + SFM, SFM + ICT, and ICT + CC, depending on their interconnections. The analysis of the heat map affiliation matrix revealed several interesting patterns and interconnections between the SDTs, highlighting the complex and interconnected nature of the SDG Framework. For example, Targets 6.6 and 12.2 have a stronger bond in CC and SFM but not in ICT, while other top SDTs are connected to the three domains. The pattern in the table also suggests that out of 733 interconnections, most SDTs are strongly connected with CC, with 413 interconnections, followed by ICT, with 192 interconnections, and SFM, respectively, with 128 interconnections.

The heat map affiliation matrix in Table 5 provides a comprehensive analysis of the interconnections between SDTs and CSI domains and clusters. It highlights the critical targets that are strongly connected in all three domains, as well as those that are connected in two domains. The matrix reveals interesting patterns and interconnections between the SDTs, providing insights into the complex and interconnected nature of the SDG Framework.

### 3.4. Network Analysis and Visualization Using Systems Thinking Approach

In the network visualization using a systems thinking approach, the affiliation matrix (Table 3) was utilized to map the interconnections between SDTs and the focal areas, CC, SFM, and ICT. The matrix visually represents these relationships, revealing patterns, overlaps, gaps, and emergent properties. Based on the matrix, SDTs were clustered into four categories, including CC + SFM, SFM + ICT, ICT + CC, and CC + SFM + ICT, based on their interconnections with the focal areas. This visualization method helps to identify patterns, overlaps, and gaps in the relationships between SDTs and the focal areas, identifying potential leverage points for addressing sustainability challenges and areas for further research and action.

Figure 7 showed the simplified interconnections of SDG targets in the three domains, namely, CC, SFM and ICT. The study found that out of 169 SDTs, 124 were interconnected with CSI Nexus, with the width of the edges depending on the number of connections ranging from 1 to 4 lines. The network has 127 nodes in total, with 124 SDTs as nodes and the three domains of CC, SFM, and ICT as network hubs. The thickness of the lines and the size of the nodes in the network visualization indicate the number of connections, with thicker lines and larger nodes representing more significant SDTs. The results revealed that the network was distributed into four CSI clusters, which are indicated by different colors: 16 SDTs shaded in purple were connected to the CC + SFM cluster, indicating that these targets are directly related to both climate change and sustainable forest management; one SDT colored in yellow was connected to the SFM + ICT cluster, suggesting that this target is closely related to both sustainable forest management and information and communication technology; 51 SDTs shaded with red were connected to the ICT + CC cluster, indicating that this targets are closely related to both ICT and climate change; and 56 SDTs shaded with pink were connected to the CC + SFM + ICT cluster, suggesting that these targets are interconnected with all three domains of CSI Nexus. These findings highlight the complex and interconnected nature of the SDG Framework, with certain SDTs closely related to specific CSI domains and others more interconnected with multiple domains. These findings suggest that some SDTs are closely related to specific CSI domains, while others are more interconnected with multiple domains, highlighting the need for integrated and coordinated approaches to address sustainability challenges.

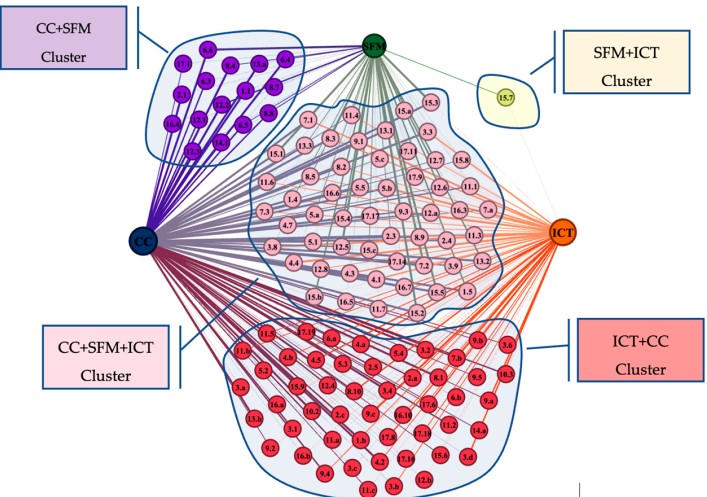

**Figure 7.** Network Visualization of 124 SDTs connected with CSI Nexus Clusters.

### 3.5. Network Analysis of Top Significant SDTs

Table 6 provides a detailed overview of the social network analysis of CSI and SDTs, focusing on the description, weighted degree, closeness centrality, betweenness centrality, and eigenvector centrality of each target and the three domains (CC, ICT, and SFM). The results reveal that the three domains, CC, ICT, and SFM, have the highest weighted degree

and closeness centrality, indicating that they are the most interconnected targets in the network. Additionally, the targets with the highest weighted degree are 7.2 and 8.3, suggesting that they have the most connections with other targets in the network.

**Table 6.** CSI and the Most Significant SDTs Using Weighted Degree Centrality Measures.

| CSI and SDTs | Description | Weighted Degree | Closeness Centrality | Betweenness Centrality | Eigenvector Centrality |
|---|---|---|---|---|---|
| CC | Climate Change | 328 | 0.9692 | 0.5005 | 1.0000 |
| ICT | Information and Communication Technology | 179 | 0.7875 | 0.3404 | 0.9048 |
| SFM | Sustainable Forest Management | 128 | 0.5478 | 0.1360 | 0.6366 |
| 7.2 | Increase global percentage of renewable energy | 10 | 0.5060 | 0.0001 | 0.1554 |
| 8.3 | Promote policies to support job creation and growing enterprises | 10 | 0.5060 | 0.0001 | 0.1554 |
| 1.4 | Equal rights to ownership, basic services, technology and economic resources | 9 | 0.5060 | 0.0001 | 0.1554 |
| 5.5 | Ensure full participation in leadership and decision-making | 9 | 0.5060 | 0.0001 | 0.1554 |
| 5.b | Promote empowerment of women through technology | 9 | 0.5060 | 0.0001 | 0.1554 |
| 7.1 | Universal access to modern energy | 9 | 0.5060 | 0.0001 | 0.1554 |
| 15.1 | Conserve and restore terrestrial and freshwater ecosystems | 9 | 0.5060 | 0.0001 | 0.1554 |
| 15.2 | End deforestation and restore degraded forests | 9 | 0.5060 | 0.0001 | 0.1554 |
| 15.3 | End desertification and restore degraded land | 9 | 0.5060 | 0.0001 | 0.1554 |
| 15.5 | Protect biodiversity and natural habitats | 9 | 0.5060 | 0.0001 | 0.1554 |
| 1.5 | Build resilience to environmental, economic and social disasters | 8 | 0.5060 | 0.0001 | 0.1554 |
| 2.3 | Double the productivity and incomes of small-scale food producers | 8 | 0.5060 | 0.0001 | 0.1554 |
| 3.3 | Fight communicable diseases | 8 | 0.5060 | 0.0001 | 0.1554 |
| 4.1 | Free primary and secondary education | 8 | 0.5060 | 0.0001 | 0.1554 |
| 6.6 | Protect and restore water-related ecosystems | 8 | 0.5020 | 0.0001 | 0.1001 |
| 8.5 | Full employment and decent work with equal pay | 8 | 0.5060 | 0.0001 | 0.1554 |
| 9.1 | Develop sustainable, resilient and inclusive infrastructures | 8 | 0.5060 | 0.0001 | 0.1554 |
| 12.5 | Substantially reduce waste generation | 8 | 0.5060 | 0.0001 | 0.1554 |
| 13.1 | Strengthen resilience and adaptive capacity to climate-related disasters | 8 | 0.5060 | 0.0001 | 0.1554 |
| 13.2 | Integrate climate change measures into policy and planning | 8 | 0.5060 | 0.0001 | 0.1554 |
| 15.4 | Ensure conservation of mountain ecosystems | 8 | 0.5060 | 0.0001 | 0.1554 |
| 4.4 | Increase the number of people with relevant skills for financial success | 7 | 0.5060 | 0.0001 | 0.1554 |
| 5.a | Equal rights to economic resources, property ownership and financial services | 7 | 0.5060 | 0.0001 | 0.1554 |
| 9.3 | Increase access to financial services and markets | 7 | 0.5060 | 0.0001 | 0.1554 |
| 12.2 | Sustainable management and use of natural resources | 7 | 0.5020 | 0.0001 | 0.1001 |
| 13.3 | Build knowledge and capacity to meet climate change | 7 | 0.5060 | 0.0001 | 0.1554 |
| 16.7 | Ensure responsive, inclusive and representative decision-making | 7 | 0.5060 | 0.0001 | 0.1554 |

Note: Descriptions of the Targets were adapted from https://sdg-tracker.org accessed on 5 February 2023.

Moreover, ICT and CC have the highest eigenvector centrality, indicating their stronger influence in the network. These findings suggest that targets related to renewable energy, job creation, equal rights, full participation, and universal access to modern energy have a high degree of connectivity in the network. The table is a part of the larger network analysis of CSI and SDTs that includes 169 SDTs generated in the statistics of Gephi. Overall, these results imply that the CSI and SDTs network is highly interconnected, and policy responses related to the integration of CSI and SDTs should consider these findings to improve their effectiveness.

The remaining SDTs have lower centrality measures, indicating that they are less connected and less central in the network. These results suggest that policy responses related to the integration of CSI and SDTs should consider the high connectivity and centrality of CC, ICT, and SFM to improve their effectiveness.

### 3.6. Quantification of Relationships between and among CC, SFM, ICT, and SDTs

Phase 3 of the methodology deals with the quantification of relationships between and among CC, SFM, ICT, and SDTs through network analysis, frequency distribution analysis, Spearman's rho correlation analysis, and hypothesis testing. The resulting output of this study is the development of CSI Nexus and SDTs Integration Framework.

3.6.1. Descriptive Analysis and Comparison of the Frequency Distributions of CC, SFM, and ICT

We provide a descriptive analysis of the frequency distributions of the strength of connections between the 169 SDTS and the three key domains: CC, SFM, and ICT. Our findings are illustrated in Figure 8, which presents the percentage distribution of these connections. Upon examining the data, it becomes evident that the majority of the SDG targets had strong and very strong connections to climate change. In fact, 23% of the targets were strongly connected to CC, while 28% were very strongly connected. Thus, a total of 51% of the SDG targets demonstrated significant connections to climate change.

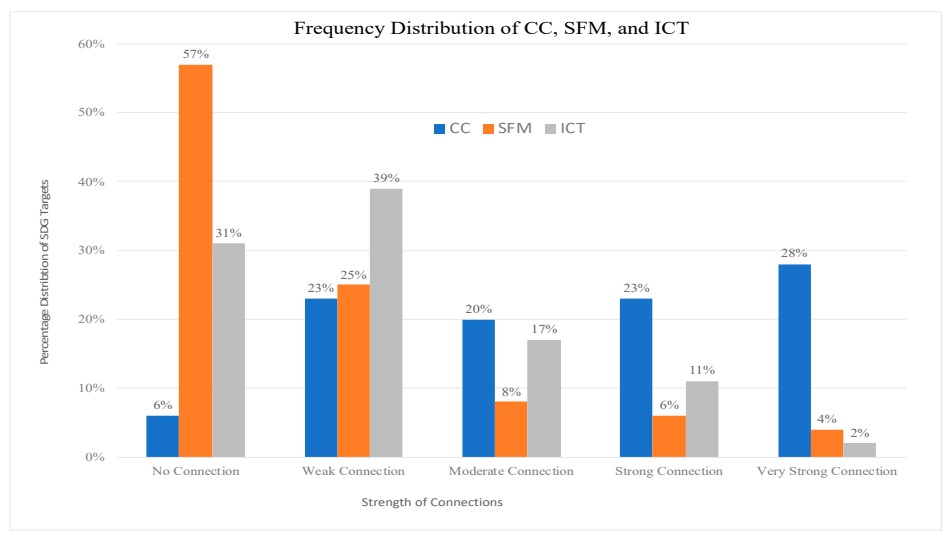

**Figure 8.** Frequency distribution of SDG targets to CC, SFM, and ICT.

In contrast, the connections between the SDG targets and Sustainable Forest Management were less pronounced. Only 6% of the targets were strongly connected to SFM, and a mere 4% were very strongly connected. This indicates that just 10% of the SDG targets had strong or very strong connections to Sustainable Forest Management. Furthermore, our analysis revealed that a significant proportion of the SDG targets (57%) had no connection to SFM at all. Similarly, the Information and Communication Technologies dimension also showed limited connections with the SDG targets. Only 11% of the targets were strongly connected to ICT, and 2% were very strongly connected. As a result, just 13% of

the SDG targets had strong or very strong connections to Information and Communication Technologies.

In summary, our descriptive analysis and comparison of the frequency distributions highlight that the majority of SDG targets are strongly or very strongly connected to climate change, whereas the connections to Sustainable Forest Management and Information and Communication Technologies are considerably weaker. This suggests that the focus of the SDG targets is predominantly on addressing climate change, while the other two dimensions may require more attention for a balanced and comprehensive approach to sustainable development.

3.6.2. Frequency Distributions Analysis

The primary objective of the frequency distribution analysis was to assess the occurrence of each SDT concerning CC, SFM, and ICT. Table 7 presents the descriptive statistics and frequency distribution analyses for the SDG targets (SDTs) aligned to CC, SFM, and ICT. In Table 7a, CC has a mean score of 2.44, suggesting a moderate to high association with SDTs on average. SFM has a mean score of 0.76, indicating a poor link, whereas ICT has a mean score of 1.14, indicating a weak to moderate relationship. The standard deviations (1.28, 1.10, and 1.05, respectively) for CC, SFM, and ICT demonstrate various degrees of link strength between SDTs and each context.

**Table 7.** Frequency Distribution SDTs in the Context of CC, SFM, and ICT.

| Descriptive Statistical Analysis of CC, SFM, and ICT | | | | Frequency Distribution of Climate Change | | | |
|---|---|---|---|---|---|---|---|
| | CC | SFM | ICT | Climate Change (CC) | Counts | % of Total | Cumulative % |
| N | 169 | 169 | 169 | No Connection | 10 | 5.9 % | 5.9 % |
| Missing | 0 | 0 | 0 | Weak Connection | 39 | 23.1 % | 29.0 % |
| Mean | 2.44 | 0.76 | 1.14 | Moderate Connection | 34 | 20.1 % | 49.1 % |
| Median | 3.00 | 0.00 | 1.00 | Strong Connection | 38 | 22.5 % | 71.6 % |
| Mode | 4.00 | 0.00 | 1.00 | Very Strong Connection | 48 | 28.4 % | 100.0 % |
| Standard deviation | 1.28 | 1.10 | 1.05 | | | | |
| Minimum | 0 | 0 | 0 | | | | |
| Maximum | 4 | 4 | 4 | | | | |
| (a) | | | | (b) | | | |

| Frequency Distribution of Sustainable Forest Management | | | | Frequency Distribution of Information and Communication Technology | | | |
|---|---|---|---|---|---|---|---|
| Sustainable Forest Management (SFM) | Counts | % of Total | Cumulative % | Information and Communication Technology (ICT) | Counts | % of Total | Cumulative % |
| No Connection | 96 | 56.8 % | 56.8 % | No Connection | 53 | 31.4 % | 31.4 % |
| Weak Connection | 42 | 24.9 % | 81.7 % | Weak Connection | 66 | 39.1 % | 70.4 % |
| Moderate Connection | 14 | 8.3 % | 89.9 % | Moderate Connection | 28 | 16.6 % | 87.0 % |
| Strong Connection | 10 | 5.9 % | 95.9 % | Strong Connection | 18 | 10.7 % | 97.6 % |
| Very Strong Connection | 7 | 4.1 % | 100.0 % | Very Strong Connection | 4 | 2.4 % | 100.0 % |
| (c) | | | | (d) | | | |

Table 7b shows that most SDTs are moderately to strongly connected to CC. 10 (5.9%) of the 169 SDTs had no connection, 39 (23.1%) had a weak connection, 34 (20.1%) had a moderate connection, 38 (22.5%) had a strong connection, and 48 (28.4%) had a very strong connection. Most SDTs have a moderate to very strong connection to CC, emphasizing the need to prioritize these objectives when formulating climate-related policies and initiatives.

The frequency distribution analysis for SFM (Table 7c) shows varied links between the 169 SDTs and SFM. 96 (56.8%) SDTs had no relationship to SFM, 42 (24.9%) had a weak connection, 14 (8.3%) had a moderate connection, 10 (5.9%) had a strong connection, and 7 (4.1%) had a very strong connection. SDTs are less connected to SFM than CC, indicating that fewer SDTs have a moderate to very strong link to SFM than CC. This stresses the need

to prioritize these objectives when establishing policies and actions to promote sustainable forest management.

The ICT frequency distribution study (Table 7d) shows that most of the 169 SDTs have weak (39.1%) or no (31.4%) links to ICT. Moderate (16.6%), strong (10.7%), and very strong (2.4%) links are less common. This demonstrates that a significant proportion of SDTs have moderate, strong, or very strong links to ICT. These findings highlight the importance of considering ICT when formulating policies and initiatives related to sustainable development.

### 3.6.3. Spearman's Rho Correlation Analysis and Hypothesis Testing

Figure 9 presents Spearman's rho Correlations Matrix for the relationships between and among CC, SFM, and ICT. This matrix visually represents the densities, statistics, and significant correlations among these three key domains.

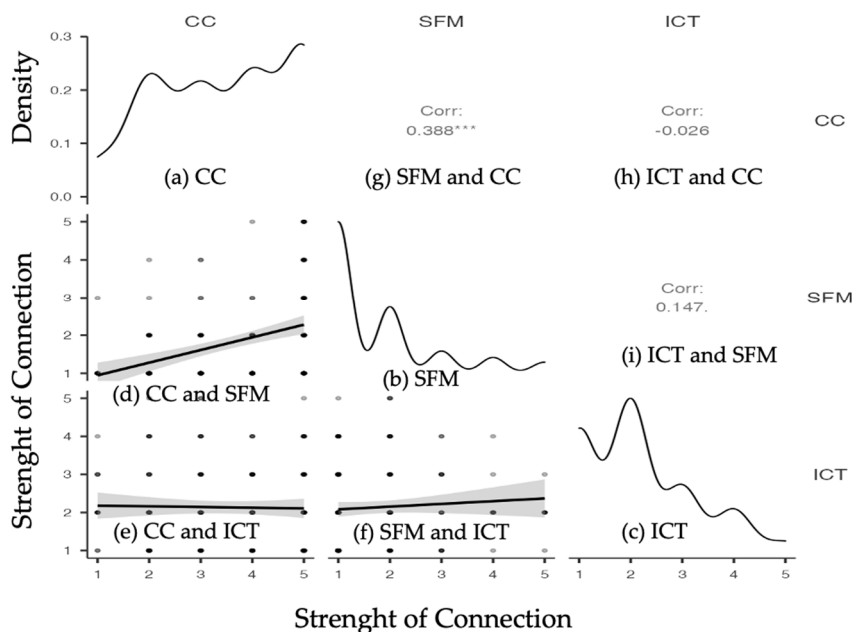

**Figure 9.** Correlation Matrix Plot of CC, SFM, and ICT. Note: *** $p < 0.001$; The correlation matrix plot shows the relationship of three variables, CC, SFM, and ICT, represented by the density curve, the scatter plot, and the correlation coefficient: (**a**) Density curve of Climate Change (CC), (**b**) Density curve of Sustainable Forest Management (SFM), (**c**) Density curve of Information and Communication Technology (ICT); Visual relationship of two variable using dots and best-fit line: (**d**) scatter plot of CC and SFM, (**e**) scatter plot of CC and ICT, and (**f**) scatter plot of SFM and ICT; The correlation coefficient of: (**g**) SFM and CC, (**h**) ICT and CC, and (**i**) ICT and SFM. The value for the *x* and *y*-axis represents the strength of connections, with values ranging from 1 (no connection), 2 (weak connection), 3 (moderate connection), 4 (strong connection), and 5 (very strong connection).

CC demonstrates a statistically significant positive association with SFM (rho = +0.388, $p < 0.001$), suggesting that as CC connections strengthen, so do SFM connections. CC does not correlate with ICT (rho = $-0.026$, $p = 0.736$), indicating they are not directly associated. ICT also has a positive association with SFM; however, the link is less (rho = 0.147, $p = 0.057$).

The density curves in Figure 7a–c show that most (SDTs are substantially linked to CC, whereas just a few are strongly linked to SFM and ICT. This shows that sustainable development strategies must address climate change.

Linear scatter plots (d–f) are used to visualize the relationships between pairs of dimensions, with each dot representing an SDT. The slope of the line of best fit for CC and SFM shows an increasing trend, indicating a positive correlation between these dimensions.

In contrast, the scatter plot for CC and ICT reveals a decreasing trend, suggesting a negative correlation. Lastly, the scatter plot for SFM and ICT displays a slightly increasing trend, indicating a weak positive correlation between these two components.

Additionally, Figure 9 highlights the complex relationships among CC, SFM, and ICT, emphasizing the need for a comprehensive understanding of these connections when developing policies and initiatives aimed at achieving sustainable development. The correlations between these domains can inform targeted actions that address the unique challenges and opportunities presented by each domain, ultimately contributing to a more balanced and effective approach to sustainable development. A single asterisk (* $p < 0.05$) denotes statistical significance with a confidence level of 95 percent. The double asterisk (** $p < 0.01$) indicates statistical significance with a degree of confidence of 99 percent, while the triple asterisk (*** $p < 0.001$) indicates statistical significance with a level of confidence of 99.9 percent.

Finally, utilizing the Spearman's rho correlation analysis results from Figure 9, we evaluate the significant association of CC, SFM, and ICT using the correlation coefficients (rho) and *p*-values for the three hypotheses:

**Hypothesis 1 (H1)** *Correlation between CC and SFM: rho = +0.388, p < 0.001; Since p < 0.05, reject $H0_1$: There is a significant positive association between CC and SFM.*

**Hypothesis 2 (H2)** *Correlation between SFM and ICT: rho = 0.147, p = 0.057; Since p > 0.05, accept $H0_2$: There is no significant association between SFM and ICT.*

**Hypothesis 3 (H3)** *Correlation between ICT and CC: rho = −0.026, p = 0.736; Since p > 0.05, accept $H0_3$: There is no significant association between ICT and CC.*

The hypothesis testing results indicate a significant positive association between Climate Change (CC) and Sustainable Forest Management (SFM). In contrast, no significant associations were found between ICT and CC and between SFM and ICT. These findings contribute to a better understanding of the interrelationships among CC, SFM, and ICT in the context of sustainable development goals, informing the development of more targeted and effective policies and initiatives.

3.6.4. CSI-Nexus and the SDG 2030 Integration Framework

Figure 10 presents a comprehensive and simplified overview of the CSI-Nexus and SDG 2030 integration framework, which incorporates Climate Change (CC), Sustainable Forest Management (SFM), Information and Communication Technology (ICT), and Sustainable Development Targets (SDTs). This Framework highlights the crucial connections and correlations between CC, SFM, and ICT, as well as the SDTs associated with these elements.

A notable finding from the Framework is the strong associations and positive correlations between CC and SFM, with 16 SDTs directly connected within the CC + SFM cluster of the CSI Nexus. This indicates that addressing climate change and promoting sustainable forest management practices are interconnected goals that contribute significantly to achieving multiple SDTs.

In contrast, SFM and ICT exhibit a positive correlation but lack a significant association. Only one SDT is directly connected to the SFM + ICT cluster, suggesting that while there is a relationship between sustainable forest management and ICT, it is not as substantial as the connection between CC and SFM.

The relationship between ICT and CC is more complex, as they are negatively correlated but without a significant association. Within the ICT + CC cluster, there are 51 SDTs directly connected, demonstrating the importance of considering the interplay between technology and climate change in achieving sustainable development goals.

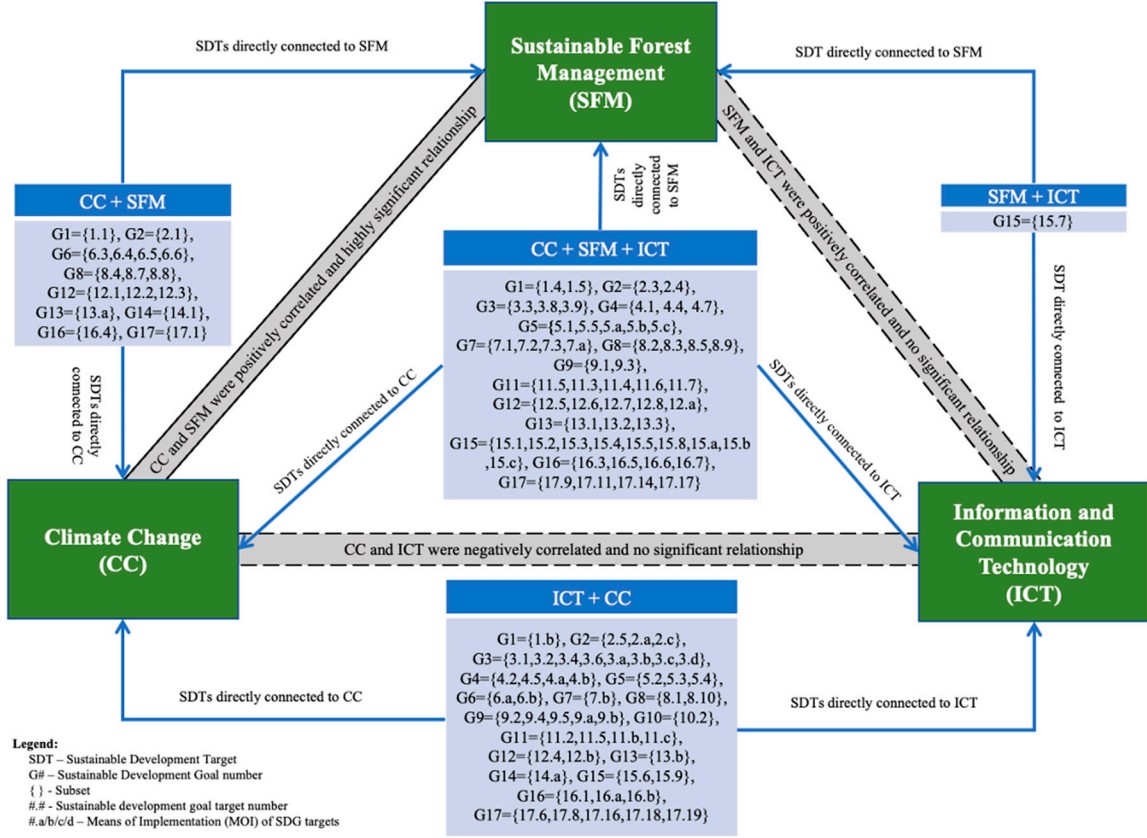

**Figure 10.** CSI Nexus and SDG 2030 Integration Framework.

Despite ICT having no significant association with CC and SFM, it still plays a vital role in the CSI Nexus. 56 SDTs are strongly and directly connected within the CC + SFM + ICT cluster, emphasizing the need for a comprehensive approach that integrates all three components to address global challenges effectively.

The various groupings of SDTs in each cluster represent a subset of the 17 Sustainable Development Goals (SDGs). For example, the notation G1 = {1.4, 1.5} indicates that in SDG 1—No Poverty, two targets are aligned with the CC + SFM + ICT cluster. Additionally, one target (G1 = {1.1}) is connected to the CC + SFM cluster, while another target (G1 = {1.b}) is connected to the ICT + CC cluster.

## 4. Discussion

In this section, we examine the interrelationships among CC, SFM, and ICT within the SDGs context. Our comprehensive methodology integrates simplified meta-data analysis, systems thinking, network analysis and visualization, and various statistical techniques to develop the CSI Nexus and SDTs Integration Framework, which informs decision-making for achieving the SDGs by 2030. Our results reveal the strong interconnectedness of the integrated CC, SFM, and ICT with 124 SDTs connected within the CSI Nexus network and the importance of context-specific policy responses. Notably, we found a significant positive association between CC and SFM, while no significant associations emerged between ICT and CC or between SFM and ICT.

### 4.1. Interconnectedness of SDTs with CC, SFM, and ICT in the Context of the CSI Nexus Clusters

The results in Figure 5 emphasize the necessity of integrating Climate Change (CC), Sustainable Forest Management (SFM), and Information and Communication Technology (ICT) for the achievement of Sustainable Development Goals (SDGs) [3]. The considerable proportion of SDTs (33.1%) within the CC + SFM + ICT cluster demonstrates the importance

of a comprehensive, interdisciplinary approach to address the interconnected challenges in these areas [4]. The notable presence of the ICT + CC cluster (30.2%) underlines the significance of technology in climate change mitigation and adaptation efforts. Policymakers should prioritize investment in ICT infrastructure and research to support sustainability [32]. The existence of the SFM + ICT and No Connection clusters (0.6% each) suggests potential for improvement in integrating SFM and ICT into broader climate change and sustainable development initiatives [58]. In conclusion, the distribution of SDTs across CSI clusters highlights the importance of considering the interconnections among CC, SFM, and ICT when pursuing the SDGs [3].

Moreover, Figure 6 revealed the clustering of literature sources connecting SDTs to CSI Nexus Clusters. It emphasizes the importance of considering the interconnected nature of CC, SFM, and ICT when pursuing the SDGs [3]. The significant role of CC4 [59] in the literature accentuates the need to address climate change as a central aspect of sustainable development. The representation of SFM3 [31] and SFM4 [33] highlights the importance of forest management practices, while ICT2 [52] emphasizes the potential contribution of technology to sustainable development. In conclusion, achieving the SDGs necessitates a holistic, integrated approach considering the interconnectedness of CC, SFM, and ICT [1].

Lastly, Table 4 revealed the frequency distributions and connections of SDTs to CSI Nexus clusters. Most SDTS are connected to CC, followed by ICT and SFM, underlining their critical role in achieving the SDGs [1]. As demonstrated by the 56 SDTs with moderate to strong connections to CC, SFM, and ICT, the strong interrelationships among CSI domains emphasize the need for integrated approaches to address global challenges [2,5]. Policymakers should consider these interrelationships when formulating sustainable development strategies [3,7]. Integrating climate change adaptation and mitigation in SFM practices can yield multiple benefits [50], while leveraging ICT can aid in managing natural resources and enhancing community resilience [48]. In conclusion, the significant connections between SDTs and the CC, SFM, and ICT domains call for an integrated approach to achieving the SDGs [3].

### 4.2. Systems Thinking Approach and Network Analysis of Top SDTs

The systems thinking approach, using a heat map affiliation matrix (Table 5) and network visualization (Figure 7), offers valuable insights into the interconnectivity between Sustainable Development Targets (SDTs) and domains, such as Climate Change (CC), Sustainable Forest Management (SFM), and Information and Communication Technology (ICT) [60]. Table 6 presents the CSI Domains and the top SDTs using degree centrality metrics. The analysis identified top significant SDTs, including Target 7.2-increase the global percentage of renewable energy, and Target 8.3-promote policies to support job creation and growing enterprises, which have strong connections across all three domains, highlighting the interconnected nature of the SDG framework and the need for addressing these targets to achieve maximum impact.

The majority of SDTs were found to be strongly connected with CC, followed by ICT and SFM, emphasizing the centrality of climate change in sustainable development and the critical role of ICT in addressing global challenges [61]. Efforts to achieve the SDGs must prioritize CC and leverage ICT as a tool to support other SDGs' implementation.

Network visualization revealed that out of 169 SDTs, 124 had strong connections with the CSI-Nexus, distributed across four distinct clusters, demonstrating the varying degrees of interconnectivity between SDTs and focal areas [62]. Understanding these interconnections allows policymakers and stakeholders to prioritize targets and interventions with the most significant potential for impact across multiple domains, increasing the likelihood of successfully achieving Sustainable Development Goals (SDGs) [63].

Furthermore, network analysis using degree centrality metrics [35] identified CC, ICT, and SFM as central players in the network, with the highest weighted degree and closeness centrality. Targets 7.2 and 8.3 exhibited the highest weighted degree, indicating strong connections with other targets. This finding suggests that by focusing on these

highly connected targets, policymakers and stakeholders can potentially address multiple challenges simultaneously, leading to synergistic outcomes [64].

### 4.3. Assessing Relationships among CC, SFM, and ICT for Sustainable Development

The analysis of frequency distributions of CC, SFM, and ICT connections to the 169 SDG targets reveals the centrality of CC in the SDG framework [5,65]. However, weaker connections with SFM and ICT highlight the need for a balanced and integrated approach to achieving the SDGs [51,66].

Given the critical role of forests in climate change mitigation, biodiversity conservation, and ecosystem services provision [67], integrating SFM into the SDG framework could enhance synergies between climate change mitigation and other goals [68]. Additionally, stronger ICT connections could harness technology's transformative potential in addressing global challenges like data-driven decision-making, remote monitoring, and innovative resource management solutions [69,70].

The frequency distribution analysis in Table 7 calls for a comprehensive approach to sustainable development that addresses disparities in connections between the SDG targets and the three key domains [1]. Prioritizing CC, SFM, and ICT is essential for a holistic and effective strategy for achieving Sustainable Development Goals [50,71].

Spearman's rho correlation analysis in Figure 9 reveals a significant positive association between CC and SFM (rho = +0.388, $p < 0.001$), suggesting potential synergies when integrating CC and SFM considerations in policymaking (Figure 9). However, the lack of significant associations between CC and ICT (rho = $-0.026$, $p = 0.736$) and between SFM and ICT (rho = 0.147, $p = 0.057$) underlines the need for further research on ICT's role in supporting other domains in the SDG context.

In conclusion, understanding the interrelationships among CC, SFM, and ICT is crucial for developing targeted and effective policies and initiatives for sustainable development (Author, Year; Nilsson et al., 2016). This knowledge can inform a more balanced approach by helping policymakers and stakeholders identify opportunities for synergistic actions addressing challenges and opportunities presented by each domain.

### 4.4. CSI Nexus and SDG 2030 Integration Framework

Figure 10 demonstrates the CSI-Nexus and SDG 2030 Integration Framework, showcasing connections among 124 SDTs within four clusters: CC + SFM, SFM + ICT, ICT + CC, and CC + SFM + ICT. CC and SFM have a strong positive correlation, sharing 16 interconnected SDTs in the CC + SFM cluster, while SFM and ICT are linked through one SDT in the SFM + ICT cluster. Although there is no significant association between ICT and CC or SFM, ICT remains substantially connected due to its presence in three clusters.

The CSI Nexus Framework recommends prioritizing the 14 goals and 56 SDTs within the CC + SFM + ICT cluster for CC, SFM, and ICT stakeholders (Author, Year). Institutions addressing these domains should focus policies and resource allocation on identified targets to ensure policy coherence and prioritize Agenda 2030 implementation for sustainable development [46,72]. Integrating CC, SFM, and ICT logic is crucial for achieving SDGs by promoting policy coherence, stakeholder engagement, and institutional coordination [73]. Inclusive stakeholder participation and coordinated efforts lead to well-informed policies and effective resource allocation, addressing challenges such as conflicting priorities and limited awareness [35].

The CSI Nexus Integration Framework outlines SDG targets, merges various logics, and fosters collaboration among CC, SFM, and ICT stakeholders, emphasizing policy consistency, stakeholder participation, and institutional coordination, offering a viable path for successful SDG integration.

In conclusion, the CSI Nexus and SDG 2030 Integration Framework in Figure 10 provide valuable insights into the connections and correlations among CC, SFM, and ICT, which can inform integrated approaches to achieving the Sustainable Development Goals.

## 5. Conclusions and Recommendations

In conclusion, this study explores the interconnections between climate change (CC), sustainable forest management (SFM), and information and communication technology (ICT) within the Sustainable Development Goals (SDG) framework. Utilizing network analysis and visualization techniques, this research identifies crucial SDG targets related to CC, SFM, and ICT and evaluates their significant relationships.

The findings reveal an imbalanced distribution of SDG targets, with CC receiving greater priority than SFM and ICT. This imbalance raises concerns about the SDG framework's effectiveness in comprehensively addressing climate change and sustainable development. The results emphasize the importance of strengthening the connections between CC, SFM, and ICT within the SDG framework to effectively address climate change adaptation and mitigation and ensure equal consideration for all three components in decision-making and policy processes.

Furthermore, the study assesses the significant relationships between CC, SFM, and ICT. A strong link exists between CC and SFM, but neither CC nor SFM shares a strong connection with ICT. This finding raises concerns that ICT might be underutilized in efforts to combat and adapt to climate change and in supporting sustainable forest management.

The CSI-Nexus and SDG 2030 Integration Framework illustrate the interrelated nature of the SDG goals and demonstrate the necessity for the systemic and contextual implementation of Agenda 2030. Policymakers, practitioners, and stakeholders in the climate change, forestry, and information and communication technology (ICT) sectors should concentrate on the established goals to ensure policy consistency and efficient resource allocation. Failing to do so may result in poor resource utilization, missed collaboration opportunities, and slower progress toward meeting the SDGs.

Overall, this study highlights the importance of integrating various aspects of sustainable development to address complex global challenges, such as climate change. The concerns raised by the findings call for increased focus on the roles of ICT and SFM in climate change mitigation and adaptation efforts. Future research should investigate new sustainability frameworks, the part of public-private partnerships in harnessing ICT for sustainable development, and the potential of ICT to address a broader range of sustainability issues.

Based on the findings of this study, the following recommendations are proposed:

1. Enhance the Sustainable Development Goals (SDG) framework by prioritizing and integrating Climate Change (CC), Sustainable Forest Management (SFM), and Information and Communication Technology (ICT) to address global sustainability challenges in a balanced manner.
2. Encourage policymakers and stakeholders to allocate equal consideration and funding to CC, SFM, and ICT within decision-making and policy processes to maximize climate change adaptation and mitigation strategies.
3. Increase investment in ICT infrastructure and research to bolster its role in supporting CC and SFM and explore public-private partnerships to leverage ICT for sustainable development.
4. Foster interdisciplinary collaboration among stakeholders in climate change, forestry, and ICT sectors to develop innovative solutions that address the interconnected nature of the SDGs and optimize resource utilization.
5. Design and evaluate new sustainability frameworks that integrate CC, SFM, and ICT, focusing on their interconnectedness and potential synergies. Conduct further research on the capacity of ICT to address broader sustainability issues and contribute to overall progress towards the 2030 Agenda.
6. Promote holistic, sustainable development by facilitating capacity building, education, and training programs on the linkages between CC, SFM, and ICT.

These recommendations seek to help establish focused and successful policies and activities that address the complex interrelationships between CC, SFM, and ICT, contributing to the 2030 Sustainable Development Goals.

**Supplementary Materials:** The following supporting information can be downloaded at: https://www.mdpi.com/article/10.3390/su15086712/s1, Figure S1: Network Visualization of Connecting CC1 to CC4, SFM1 to SFM4, ICT1 to ICT4, and the SDTs into One Network System; Table S1: List of Connected SDTs Data Source Aligned to CC, SFM, and ICT; Table S2: Affiliation Matrix Mapping of CC, SFM, ICT, SDTs, SDGs, and CSI-Nexus Clusters; Table S3: Network Centrality Measures of SDTs and the CSI Clusters with Corresponding Descriptions

**Author Contributions:** Conceptualization, E.E.E., R.F.A.J., A.S. and R.S.P.; methodology, E.E.E., R.F.A.J. and A.S.; software, E.E.E.; validation, R.F.A.J., A.S., R.S.P., P.D.S., R.Q.L., M.R.Y.T. and M.C.M.F.; formal analysis, E.E.E.; investigation, E.E.E.; resources, R.F.A.J., A.S., R.S.P. and R.Q.L.; data curation, E.E.E., R.F.A.J. and A.S.; writing—original draft preparation, E.E.E.; writing—review and editing, R.F.A.J. and A.S.; visualization, E.E.E.; supervision, R.F.A.J. and A.S.; project administration, R.F.A.J., A.S. and R.S.P.; funding acquisition, E.E.E., R.F.A.J., A.S. and R.S.P.; All authors have read and agreed to the published version of the manuscript.

**Funding:** This sandwich research collaboration was funded by Mindanao State University Systems and MSU-Iligan Institute of Technology, Philippines; Kastamonu University, Turkey; and the Erasmus+.

**Institutional Review Board Statement:** Not applicable.

**Informed Consent Statement:** Not applicable.

**Data Availability Statement:** The collected Data source of the alignment of 169 Sustainable Development Goal Targets are available in the following URLs: (1) https://bit.ly/3jJBlCu, accessed on 30 October 2022; (2) https://bit.ly/3vztl9P, accessed on 30 October 2022; (3) https://bit.ly/3vx2Xx7, accessed on 30 October 2022; (4) https://bit.ly/3CEeAqj, accessed on 30 October 2022; (5) https://bit.ly/3Z7W6rQ, accessed on 30 October 2022; (6) https://bit.ly/3G66RlH, accessed on 30 October 2022; (7) https://bit.ly/3X0G0y4, accessed on 30 October 2022; (8) https://bit.ly/3WBVLvC, accessed on 30 October 2022; (9) https://bit.ly/3G7MMeQ, (accessed on 30 October 2022; 10) https://bit.ly/3Q9Cxv9, accessed on 30 October 2022; (11) https://bit.ly/3WWKeGX, accessed on 30 October 2022; and (12) https://bit.ly/3i8mrFe, accessed on 30 October 2022.

**Acknowledgments:** The authors wish to thank the following: The Administrators of MSU Systems for granting me an opportunity to conduct a research collaboration at Kastamonu University, Turkey; the Administrators of MSU-Iligan Institute of Technology for giving me a three-year Faculty Development Program (FDP) for me to finish my Dissertation; administrators of Kastamonu University, Turkey, especially, to all academic and technical staff of the Faculty of Forestry, for granting me an opportunity to use their resources and offices during my six months (3 months in 2017 and 3 months in 2019) stayed to conduct my research; Venus R. Parmisana for reviewing and editing our manuscripts; to the unwavering supports and prayers of my family; and lastly, to the Lord Almighty for divine guidance and intellectual analysis to finish my Dissertation.

**Conflicts of Interest:** The authors declare no conflict of interest.

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
