# Peer review of "Climate Change, Sustainable Forest Management, ICT Nexus, and the SDG 2030: A Systems Thinking Approach"

_sustainability, doi:10.3390/su15086712_

Round 1

Reviewer 1 Report

The effort that has been made to "connect" different logics is salutary. The visual representation is successful. The modalities of their realizations are very explicit. This is one of the values of this work. One particularly appreciates this will to make visible the assumptions and the choices in the exercise of representation.

The conclusion of the article is however very disappointing. You describe the contribution of your work very badly. You do not explain the issues that arise from your results. 

You do not address the mediations that will have to be implemented to generate the relations and collaborations between the different logics that you wished to bring into conversation.

Reviewer 2 Report

Comments for Revision

1. In abstract, methodology is not clear in the abstract.

2. Introduction: It is very lengthy, try to precise it with special focus on background, problem statement, novelty and key objectives.

3. Methodology: It is complicated and unclear. It needs to be revised for better understanding of the readers.

4. In methodology it is written that objective is to investigate and identify SDG targets aligned and connected to CC, SFM, ICT. Why there is need to correlate with climate change and forest management??

5. Results: Table 3 shows frequency Distribution of SDG Targets to CC, SFM, and ICT Clusters. There is no strong connection of CC+SFM with any SDG or SDT. How is it possible? See SDG 12 (climate action).

6. Conclusion: General statement is given as conclusion.

7. At the end of conclusion, there is also no specific recommendation.

8. Manuscript needs to be revised intensively and re-organize the draft.

9. English language should be improved.

Reviewer 3 Report

in the attached file

Round 2

Reviewer 2 Report

Manuscript is acceptable for publication.